# Pyridoxamine Attenuates Doxorubicin-Induced Cardiomyopathy without Affecting Its Antitumor Effect on Rat Mammary Tumor Cells

**DOI:** 10.3390/cells13020120

**Published:** 2024-01-09

**Authors:** Sibren Haesen, Eline Verghote, Ellen Heeren, Esther Wolfs, Dorien Deluyker, Virginie Bito

**Affiliations:** UHasselt, Cardio & Organ Systems (COST), BIOMED, Agoralaan, 3590 Diepenbeek, Belgium; sibren.haesen@uhasselt.be (S.H.); everghote@hotmail.be (E.V.); ellen.heeren@uhasselt.be (E.H.); esther.wolfs@uhasselt.be (E.W.); dorien.deluyker@uhasselt.be (D.D.)

**Keywords:** breast cancer, rat mammary tumor cells, doxorubicin, cardiotoxicity, pyridoxamine

## Abstract

Doxorubicin (DOX) is commonly used in cancer treatment but associated with cardiotoxicity. Pyridoxamine (PM), a vitamin B6 derivative, could be a cardioprotectant. This study investigated the effect of PM on DOX cardiotoxicity and DOX antitumor effectiveness. Sprague Dawley rats were treated intravenously with DOX (2 mg/kg/week) or saline over eight weeks. Two other groups received PM via oral intake (1 g/L in water bottles) next to DOX or saline. Echocardiography was performed after eight weeks. PM treatment significantly attenuated the DOX-induced reduction in left ventricular ejection fraction (72 ± 2% vs. 58 ± 3% in DOX; *p* < 0.001) and increase in left ventricular end-systolic volume (0.24 ± 0.02 µL/cm^2^ vs. 0.38 ± 0.03 µL/cm^2^ in DOX; *p* < 0.0001). Additionally, LA7 tumor cells were exposed to DOX, PM, or DOX and PM for 24 h, 48 h, and 72 h. Cell viability, proliferation, cytotoxicity, and apoptosis were assessed. DOX significantly reduced LA7 cell viability and proliferation (*p* < 0.0001) and increased cytotoxicity (*p* < 0.05) and cleaved caspase-3 (*p* < 0.001). Concomitant PM treatment did not alter the DOX effect on LA7 cells. In conclusion, PM attenuated DOX-induced cardiomyopathy in vivo without affecting the antitumor effect of DOX in vitro, highlighting PM as a promising cardioprotectant for DOX-induced cardiotoxicity.

## 1. Introduction

Breast cancer is the most frequent cancer in women, with worldwide more than two million diagnoses each year [1]. Over the last few decades, cancer prognosis has improved tremendously due to earlier detection and treatment advances, resulting in an increased population of cancer survivors [2]. Approximately one third of breast cancer patients receive adjuvant anthracycline chemotherapy, including doxorubicin (DOX) [3]. Since their discovery in the 1960s, anthracyclines (e.g., DOX) have proven to be highly effective drugs for the treatment of various cancer types, including breast cancer. As a result, they have become a cornerstone of modern cancer therapy. However, DOX treatment is associated with dose-dependent cardiotoxicity, which manifests during treatment or years after treatment completion [3]. The heart damage caused by DOX resembles dilated cardiomyopathy, characterized by an enlarged left ventricle (LV), weakened heart muscle, and a decline in LV ejection fraction (LVEF), ultimately leading to heart failure (HF) [4]. At seven years post-cancer diagnosis, cardiovascular disease (CVD) mortality dominates mortality from primary cancer in older breast cancer patients [5]. Therefore, DOX-induced cardiotoxicity is a major concern for both oncologists and cardiologists. The pathophysiology of DOX-induced cardiotoxicity is multifactorial, involving oxidative stress, mitochondrial dysfunction, iron overload, and the inhibition of DNA topoisomerase II beta activity in cardiomyocytes [6]. Cardioprotection in DOX-treated cancer patients is currently suboptimal [7], which indicates that research focusing on cardioprotective strategies is necessary.

Pyridoxamine (PM) is a natural form of vitamin B6 known for its capacity to chelate metal ions and to display anti-inflammatory and antioxidant properties [8]. Previous research has shown that PM limits the progression of myocardial ischemia and diabetes-associated atherosclerosis without inducing side effects [9,10], emphasizing its high translational value. However, whether PM promotes cardioprotection for DOX-induced cardiotoxicity remains to be explored. Moreover, increasing evidence points to a negative interaction between CVD and cancer [11], which can be partially explained by circulating factors excreted by damaged cardiac tissue and cancer cells [12]. This tight interplay between both conditions and the need for effective cardioprotective strategies in DOX-induced cardiotoxicity highlight the importance of evaluating the effect of the concomitant treatment of DOX and potential cardioprotective strategies on tumor cells.

In our study, we examined (1) the effect of PM treatment in a rat model of DOX-induced cardiomyopathy, (2) the effect of PM on the antitumor efficacy of DOX on LA7 mammary tumor cells in vitro, and (3) the potential antitumor effect of PM itself. In this study, we confirmed that PM attenuates the development of DOX-induced cardiomyopathy without affecting the effect of DOX on LA7 tumor cell characteristics.

## 2. Materials and Methods

### 2.1. Animal Model of DOX-Induced Cardiomyopathy

All animal experiments followed the EU Directive 2010/63/EU for animal testing and were approved by the local ethical committee (Ethical Commission for Animal Experimentation, UHasselt, Diepenbeek, Belgium, ID 201942, ID 202154, and ID 202139K). Animals were fed a standard pellet diet and were group-housed with water available ad libitum. Female Sprague Dawley rats (six weeks old, Janvier Labs, Le Genest-Saint-Isle, France) were randomized into two groups. The first group received DOX (N = 14, 2 mg/kg intravenously, Accord Healthcare B.V., Utrecht, The Netherlands) once a week for eight weeks (16 mg/kg cumulative dose). The second group received an equal volume of 0.9% saline (CTRL; N = 14). In addition to DOX or saline, two other groups concomitantly received pyridoxamine (PM, 1 g/L pyridoxamine dihydrochloride dissolved in water, sc-219673, Santa Cruz Biotechnology, Inc., Heidelberg, Germany) ad libitum via the water bottles provided in the cage throughout the study for eight weeks (DOX+PM; N = 18, CTRL+PM; N = 14). The PM dose was based on other studies [9,13] and the administration was started from the first injection with DOX or saline. The cumulative dose of 16 mg/kg DOX was equal to the clinically relevant DOX dose of 592 mg/m^2^ and was in line with the doses used in other preclinical studies [14]. The protocol for the induction of chronic DOX cardiotoxicity was based on current guidelines on the use of preclinical models in cardio-oncology [15]. Eight weeks after treatment initiation, transthoracic echocardiography was performed to assess LV parameters. The animals were euthanized with an overdose of sodium pentobarbital (150 mg/kg intraperitoneal, Dolethal, Vetoquinol).

### 2.2. Transthoracic Echocardiography

The rats were placed in a supine position and anesthetized with 2% isoflurane anesthesia supplemented with oxygen. To minimize artifacts, the thorax was shaved and depilatory cream was applied. Transthoracic echocardiography of the LV was performed with a Vevo^®^ 3100 high-resolution imaging system and a 21 MHz MX250 transducer (FUJIFILM VisualSonics, Inc., Amsterdam, The Netherlands), as described before [16]. The vital physiological parameters (i.e., heart rate, respiratory rate, body temperature, and ECG signals) were monitored non-invasively using the accompanying Vevo Imaging Station. The LV ejection fraction (LVEF), LV cardiac output (LVCO), and end-systolic and end-diastolic volumes (LVESV and LVEDV) were measured from parasternal long-axis views in 2D B-mode (EKV), which were acquired by tracing the LV endocardial border in the end-systole and end-diastole. LVCO was normalized to the body surface area (BSA) and expressed as the LV cardiac index. LVESV and LVEDV were also normalized to BSA. Analyses were performed using Vevo^®^ LAB (Vevo^®^ LAB software, version 5.6.1, FUJIFILM VisualSonics, Inc.) by two researchers independently.

### 2.3. Cell Culture

LA7 rat mammary tumor cells were obtained from ATCC (No. CRL-2283). They were cultured in Dulbecco’s Modified Eagle Medium (DMEM) (Cat#41966029, Gibco, Thermo Fisher Scientific, Merelbeke, Belgium) with 10% fetal calf serum, 1% L-glutamine, and 1% penicillin/streptomycin. The cells were incubated at 37 °C in a humidified incubator with a 5% CO_2_ atmosphere, and the medium was changed every two days. The different concentrations of DOX (2 mg/mL, Accord Healthcare B.V.) that were added to assess the dose-dependent antitumor effect of DOX were 0.5 µg/mL, 1 µg/mL, 1.75 µg/mL, 2 µg/mL, 2.5 µg/mL, 5 µg/mL, and 10 µg/mL. The applied dose range was based on other in vitro studies and blood DOX concentrations in treated patients [17,18,19]. To study the effect of PM on DOX antitumor efficacy, the following conditions were added to the cells: 1 µg/mL DOX, 100 µM PM, and 1 µg/mL DOX with 100 µM PM. LA7 cells that were cultured in the medium were used as a negative control.

### 2.4. Cell Viability Assay

For viability assays, the cells were seeded at a density of 6.25 × 10^3^ cells/cm^2^ and allowed to adhere for 24 h before the addition of the conditions described before, based on the guidelines of the IncuCyte^®^ S3 Live-Cell Analysis System (Sartorius, Sint-Lambrechts-Woluwe, Belgium) and comparable to other studies [17,20]. To determine cell viability, an Alamar Blue assay was performed according to the manufacturer’s guidelines and other studies [21]. After 24, 48, and 72 h exposure to the different conditions, the medium of the cells was replaced with an Alamar Blue solution diluted in standard medium (1:10, cat#DAL1025, Thermo Fisher Scientific). Following 4 h incubation at 37 °C, the Alamar Blue solution with cells was transferred to a dark 96-well plate. Fluorescence was measured with a fluorescence plate reader (Clariostar Plus, BMG Labtech, Ortenberg, Germany; excitation: 570 nm, emission: 600 nm, gain: 2000). Experiments were performed in triplicate. Data were normalized to the negative control.

### 2.5. Proliferation and Cytotoxicity Assays

For proliferation and cytotoxicity assays, the cells were seeded at a density of 6.25 × 10^3^ cells/cm^2^ and 1.56 × 10^4^ cells/cm^2^, respectively, and allowed to adhere for 24 h before the addition of the conditions described before, based on the guidelines of the IncuCyte^®^ S3 Live-Cell Analysis System (Sartorius) and comparable to other studies [17,20]. After 24, 48, and 72 h exposure to the different conditions, proliferation and cytotoxicity were studied using the IncuCyte^®^ S3 Live-Cell Analysis System (Sartorius) according to the manufacturer’s guidelines and other studies [22]. For the cytotoxicity experiments, the Incucyte^®^ Cytotox Green Reagent diluted in standard medium (1:40,000, Sartorius) was used. Images were taken every two hours for three days with a 10× lens and each condition was run in triplicate. For proliferation and cytotoxicity experiments, phase images were acquired using phase contrast. For cytotoxicity experiments, fluorescent images were acquired using a green fluorescence channel with 400 ms exposure. Proliferation was expressed as % confluence. Cytotoxicity was calculated as the total Cytotox Green area (µm^2^) divided by % confluence.

### 2.6. Immunocytochemistry

LA7 cells were seeded on coverslips at a density of 5.26 × 10^4^ cells/cm^2^, which was comparable to other studies [23]. Following attachment, four different conditions were induced: 1 µg/mL DOX, 100 µM PM, 1 µg/mL DOX with 100 µM PM, and control (standard culture medium). After 24 h, the cells were fixed with 4% paraformaldehyde for 20 min, permeabilized with 0.05% Triton X100 (Merck Life science B.V., Hoeilaart, Belgium) for 30 min, and blocked for one hour with serum-free protein block (X0909, Dako, Agilent Technologies, Machelen, Belgium). To analyze apoptosis, LA7 cells were incubated overnight at 4 °C with cleaved caspase-3 antibody (1:1000, cat#9664S, Cell Signaling Technology, Leiden, The Netherlands), followed by incubation for one hour with a secondary antibody (donkey anti-rabbit Alexa Fluor 555, 1:400, cat#A21430, Thermo Fisher). All antibodies were diluted in 1× phosphate-buffered saline (PBS). Negative controls were included by omitting the primary antibody. Nuclei staining was performed with 4’,6 diamidino-2-phenylindole (DAPI, 2 µg/mL, Sigma-Aldrich, Overijse, Belgium). Slides were mounted in a fluorescent mounting medium (Fluoromount-G™, Invitrogen by Thermo Fisher Scientific). Images were acquired in five random fields per slide at 20× magnification using a Leica fluorescence microscope (DM 4000 B LED, Leica Microsystems, Diegem, Belgium) with the Leica Application Suite X software (version 4.6.0). The total fluorescence was quantified using the Fiji software (v1.53t) by correcting the integrated density by the number of cells [24]. Two operators blinded to group allocation performed the analysis independently.

### 2.7. Statistical Analysis

Statistical analysis was performed using GraphPad Prism (GraphPad Software, version 9.5.0). The normal distribution of data was assessed with the D’Agostino and Pearson normality test. For experiments performed at multiple time points, the two-way ANOVA test or mixed-effects analysis for repeated measurements with Bonferroni post hoc analysis was used. For experiments performed at a single time point, the parametric one-way ANOVA test with Bonferroni post hoc analysis or the non-parametric Kruskal–Wallis test with Dunn’s multiple comparisons analysis was used. In the case of unequal variances, the Brown–Forsythe ANOVA test with Dunnett’s T3 post hoc analysis was used. All data are expressed as the mean ± standard error of the mean (SEM). Outliers (ROUT method, Q = 1%) were excluded. *p* < 0.05 was considered statistically significant.

## 3. Results

### 3.1. PM Attenuates DOX-Induced LV Cardiomyopathy In Vivo

At baseline, no differences in the echocardiographic parameters of LV function and volumes were found between groups (Appendix A). Eight weeks after DOX initiation, LVEF was significantly reduced in DOX-treated rats compared to CTRL (58 ± 3% vs. 82 ± 1%; *p* < 0.0001) (Figure 1a). The PM treatment (DOX+PM) significantly attenuated the DOX-induced reduction in LVEF (72 ± 2%; *p* = 0.0003 vs. DOX). LVEF in DOX+PM rats remained different from CTRL rats receiving PM (CTRL+PM) (80 ± 1%; *p* = 0.0043 vs. DOX+PM), indicating partial cardioprotection by PM. The LV cardiac index was similar between DOX and CTRL rats (Figure 1b). The PM treatment in DOX rats (DOX+PM) increased the LV cardiac index (0.21 ± 0.01 mL/min/cm^2^ vs. 0.16 ± 0.02 mL/min/cm^2^ in DOX; *p* = 0.0305), indicating improved cardiac function. In line with changes in systolic LV function, an increase in LVESV/BSA was observed in DOX rats (0.38 ± 0.03 µL/cm^2^ vs. 0.14 ± 0.01 µL/cm^2^ in CTRL; *p* < 0.0001) (Figure 1c). The PM treatment in DOX rats (DOX+PM) significantly attenuated the DOX-induced increase in LVESV/BSA (0.24 ± 0.02 vs. DOX; *p* < 0.0001). DOX administration did not significantly increase LVEDV/BSA (*p* = 0.1013) (Figure 1d). LVEDV/BSA was different between CTRL rats and CTRL rats that received PM (CTRL+PM) (0.72 ± 0.03 vs. 0.85 ± 0.03; *p* = 0.0248).

### 3.2. DOX Exhibits a Dose-Dependent Antitumor Effect

Cultured LA7 mammary tumor cells showed a typical epithelial polygonal morphology and the formation of branched-like structures that recapitulated the mammary architecture (Appendix A). As shown in Figure 2a,b, increasing concentrations of DOX reduced the viability and proliferation of LA7 cells. After 24 h, DOX concentrations above 0.5 µg/mL decreased cell viability by more than 50% (−49% at 1 µg/mL DOX, −40% at 1.75 µg/mL, −39% at 2.5 µg/mL, −35% at 5 µg/mL, and −32% at 10 µg/mL) (Figure 2a). After 48 and 72 h, the viability of LA7 cells was less than 20% for all DOX concentrations. Furthermore, DOX reduced the proliferation of cells by more than 90% compared to standard culture medium after 48 and 72 h exposure (Figure 2b). In line with these findings, cytotoxicity increased with increasing DOX concentrations, further supporting the dose-dependent antitumor effect of DOX (72 h: 41 a.u. in standard culture medium, 482 a.u. in 0.5 µg/mL DOX, 594 a.u. in 1 µg/mL, 1185 a.u. in 1.75 µg/mL, 3023 a.u. in 2.5 µg/mL, 10,056 a.u. in 5 µg/mL, and 17,779 a.u. in 10 µg/mL) (Figure 2c).

### 3.3. PM Does Not Interfere with the Antitumor Effect of DOX on Mammary Tumor Cells

To examine whether PM interferes with the antitumor efficacy of DOX, LA7 cell viability and proliferation were measured after exposure to 1 µg/mL DOX, 100 µM PM, or both DOX and PM for 24, 48, and 72 h. DOX significantly reduced LA7 cell viability (all time points: *p* < 0.0001) (Figure 3a) and proliferation (24 h: *p* = 0.0005; 48 and 72 h: *p* < 0.0001) (Figure 3b). Concomitant treatment with PM did not change the viability of LA7 cells (ns for DOX+PM vs. DOX at all time points). In addition, cell proliferation was not different between LA7 cells treated with DOX and those treated with both DOX and PM (ns for DOX+PM vs. DOX at all time points). PM treatment alone significantly reduced LA7 cell viability at 24 (*p* = 0.0085) and 72 h (*p* = 0.0395) but did not change the proliferation.

### 3.4. PM Does Not Affect the Cytotoxic and Apoptotic Effect of DOX

As shown in Figure 4a, cytotoxicity was significantly increased after DOX exposure for 48 and 72 h (48 h: *p* = 0.0350; 72 h: *p* = 0.0343). Concomitant treatment with PM did not affect the cytotoxic effect of DOX at 24, 48, and 72 h (ns for DOX+PM vs. DOX at all time points). PM alone showed no cytotoxic effects. Since cytotoxicity can result in apoptosis, the cleaved caspase-3 levels were measured. Compared to controls, the cleaved caspase-3 levels were significantly increased after DOX exposure (*p* = 0.0002 (Figure 4b) but did not change upon the addition of PM (ns for DOX+PM vs. DOX), suggesting that PM does not affect DOX-induced apoptosis. PM alone did not significantly increase the cleaved caspase-3 levels.

## 4. Discussion

There is a clinical need for novel cardioprotective strategies against DOX-induced cardiotoxicity that do not interfere with DOX’s antitumor activity. In this context, this study is the first to report that PM offers cardioprotection in a rat model of DOX-induced cardiomyopathy without affecting the antitumor efficacy of DOX on tumor cells in vitro.

### 4.1. PM as a Cardioprotectant during DOX Treatment without Affecting Antitumor Efficacy

During the last few years, multiple clinical studies have investigated the use of cardiovascular drugs to prevent anthracycline cardiotoxicity, showing inconsistent findings. [25]. In addition, the clinical use of the iron chelator dexrazoxane (DRZ) has been restricted [26]. As a result, cardioprotection in DOX-treated cancer patients is currently suboptimal [7], which indicates that research focusing on other cardioprotective strategies is necessary.

In this study, we focused on PM as a potentially novel cardioprotectant, for which our research group has previously shown that its administration improves the cardiac phenotype in a rat model of myocardial ischemia [9]. The cardioprotective nature of PM is related to its anti-inflammatory and antioxidative properties and to its capacity to chelate metal ions, including iron [8]. Because DOX-induced cardiotoxicity is mediated through myocardial inflammation, increased oxidative stress, and iron overload in cardiac tissue, PM could be a potential cardioprotectant for this condition [6]. Therefore, we investigated the effect of eight weeks of PM administration in rats that were chronically treated with DOX. In this model of DOX-induced cardiomyopathy, we observed significant systolic LV dysfunction, characterized by reduced LVEF and increased LVESV. This typical phenotype is also observed in other preclinical studies [27,28] and is similar to what is clinically observed in DOX-treated cancer patients [4]. In our confirmed model of DOX-induced cardiomyopathy, we found that rats from the DOX+PM group showed improved LVEF and LVESV compared to the DOX group. These results indicate that PM limits the increase in LV volumes and the reduction in LV systolic function induced by DOX, and thus attenuates the development of DOX-induced cardiomyopathy. In a pilot study by our group, we have previously shown that pyridoxamine protects against cardiotoxicity after doxorubicin chemotherapy [29], and, in other work, we have shown that the cardioprotective mechanism of PM can be explained by its ability to reduce myocardial fibrosis and inflammation, and to restore redox and iron regulation in mitochondria, when concomitantly administered with DOX [30]. In line with our results, Moriyama et al. (2010) demonstrated cardiac protection through enhanced fractional shortening after daily PM administration in DOX-treated male rats [31]. Moreover, PM significantly suppressed oxidative stress and the subsequent DOX-induced increase in myocardial pentosidine and Nε-(carboxymethyl)lysine levels, which are advanced glycated end-products (AGEs). This suggests the involvement of AGEs in DOX cardiotoxicity and the ability of PM to inhibit AGE formation, contributing to its cardioprotective mechanisms. However, PM was administered intraperitoneally, which is not consistent with potential clinical application [31]. In humans, Van den Eynde et al. (2023) recently showed that an eight-week intervention with PM reduced endothelial dysfunction in individuals with abdominal obesity [32]. Moreover, in diabetic patients with nephropathy, the PM treatment significantly reduced the change in serum creatinine from baseline and thus improved renal function [33]. In addition, vitamin B6-containing foods such as date palm have been shown to offer protection against DOX cardiotoxicity in rats [34], although more research is needed on the cardioprotective potency of PM-containing foods against DOX-induced cardiotoxicity. Besides its cardioprotective effect, the PM treatment was safe as it did not induce any side effects, which has also been confirmed in other preclinical studies [9,10] and in clinical trials [32,33]. Altogether, these findings suggest that PM could be a promising cardioprotectant for DOX-treated cancer patients at risk of cardiotoxicity.

However, PM treatment should not affect DOX antitumor efficacy with the risk of hampering patient treatment. To investigate this, we exposed LA7 mammary tumor cells concomitantly to DOX and PM treatment and assessed multiple tumor cell characteristics. We found that PM does not change the dose-dependent effect of DOX on LA7 cell viability, proliferation, cytotoxicity, and apoptosis. This dose-dependent effect of DOX on LA7 cells aligns with other studies that have examined the antitumor effect of DOX on other breast cancer cells, such as MCF-7 and MDA-MB-231 cells [17,23,35]. In line with these findings, a clinical trial revealed that the vitamin B6 derivative pyridoxine had no impact on the antitumor effect of the chemotherapeutic drug capecitabine in patients with advanced colorectal or breast carcinoma [36]. Galluzzi et al. (2012) further supported the evidence that vitamin B6 does not affect chemotherapy efficacy in mice transplanted with Lewis lung carcinoma cells [37]. In our study, we also found that PM treatment alone has no consistent beneficial effect on tumor cell characteristics and does not possess antitumor properties itself. Consistently, Matsuo et al. (2019) showed no antiproliferative activity of PM in HepG2 hepatoma cells and MKN45 gastric cancer cells [38]. In addition, PM did not change the expression of the pro-apoptotic genes caspase-8 and BAX in SHSY5Y cells, confirming that PM does not affect apoptotic cell death [39].

### 4.2. Current Limitations and Future Perspectives

Although these findings are promising, they must be confirmed in an in vivo setting for translation to the clinic. Future work includes investigating the cardioprotective effect of PM treatment in a preclinical model of experimental cancer and DOX-induced cardiotoxicity. In addition, the underlying mechanisms of PM cardioprotection and whether PM protects against DOX-induced vascular dysfunction deserve further investigation.

## 5. Conclusions

Taken together, our data indicate that PM attenuates DOX-induced cardiomyopathy in vivo and does not affect the DOX-induced reduction in viability and proliferation and increase in cytotoxicity and apoptosis of LA7 mammary tumor cells in vitro, highlighting PM as a promising cardioprotectant for DOX-induced cardiotoxicity.

## Figures and Tables

**Figure 1 cells-13-00120-f001:**
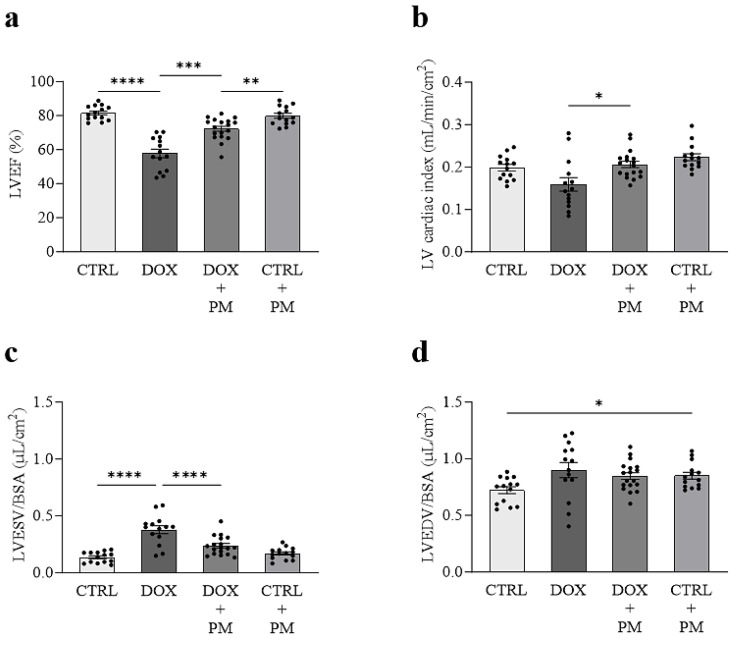
PM treatment attenuates the reduction in systolic LV function and the increase in systolic LV volume in DOX-treated rats. LVEF (**a**), LV cardiac index (**b**), LVESV (**c**), and LVEDV (**d**) measured in rats eight weeks after the initiation of treatment with saline (CTRL; N = 14), doxorubicin (DOX; N = 14), doxorubicin and pyridoxamine (DOX+PM; N = 18), and saline and pyridoxamine (CTRL+PM; N = 14). The cardiac index was calculated as cardiac output normalized to body surface area (BSA). LVESV and LVEDV were also normalized to BSA. Data are presented as mean ± SEM. * *p* < 0.05, ** *p* < 0.01, *** *p* < 0.001, and **** *p* < 0.0001. LV, left ventricle. LVEF, left ventricular ejection fraction. LVEDV, left ventricular end-diastolic volume. LVESV, left ventricular end-systolic volume.

**Figure 2 cells-13-00120-f002:**
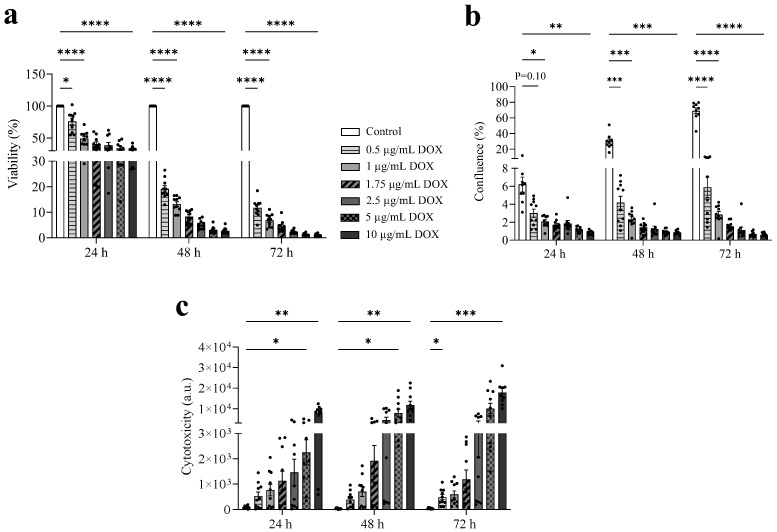
Increasing DOX concentrations reduce LA7 cell viability and proliferation and increase cytotoxicity after 24, 48, and 72 h exposure. (**a**–**c**) LA7 mammary tumor cells were exposed to different DOX concentrations (0.5, 1, 1.5, 1.75, 2.5, 5, and 10 µg/mL). Cell viability (**a**), proliferation (**b**), and cytotoxicity (**c**) were measured after 24, 48, and 72 h exposure (N = 10 repetitions/condition). LA7 cells cultured in a standard culture medium were used as a control. Data are presented as mean ± SEM. * *p* < 0.05, ** *p* < 0.01, *** *p* < 0.001, **** *p* < 0.0001. DOX, doxorubicin.

**Figure 3 cells-13-00120-f003:**
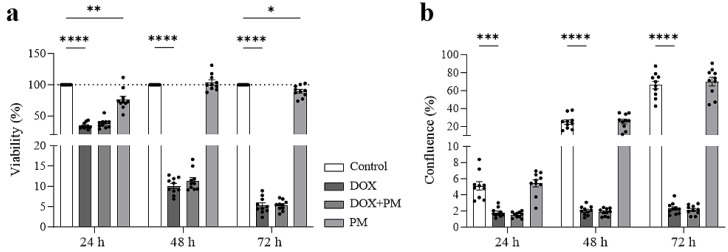
PM does not interfere with the reduced LA7 cell viability and proliferation after DOX exposure. LA7 mammary tumor cells were exposed to DOX (1 µg/mL), PM (100 µM), or both DOX and PM. Cell viability (**a**) and proliferation (**b**) were measured after 24, 48, and 72 h exposure (N = 10 repetitions/condition). Proliferation is expressed as % confluence. Data are presented as mean ± SEM. * *p* < 0.05, ** *p* < 0.01, *** *p* < 0.001, **** *p* < 0.0001. DOX, doxorubicin. PM, pyridoxamine.

**Figure 4 cells-13-00120-f004:**
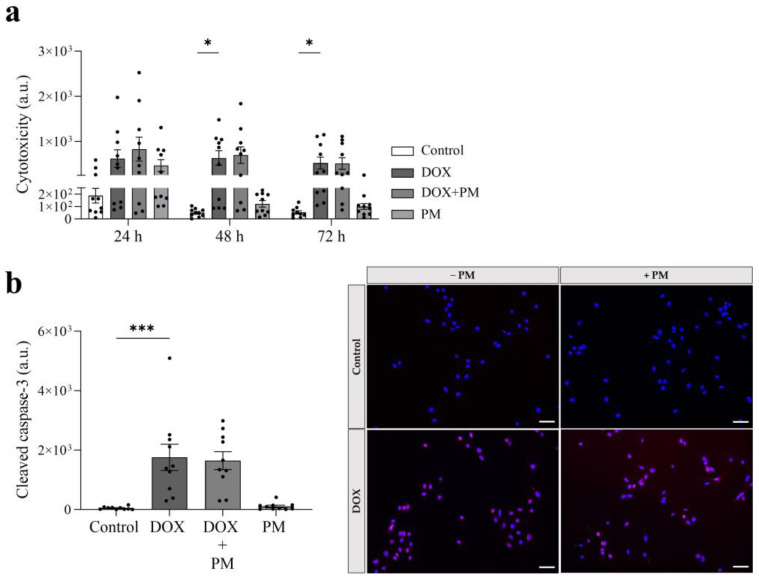
PM does not affect DOX-induced LA7 cell cytotoxicity and apoptosis. LA7 mammary tumor cells were exposed to DOX (1 µg/mL), PM (100 µM), or both DOX and PM. Cytotoxicity (**a**) was measured after 24, 48, and 72 h exposure (N = 10 repetitions/condition). Apoptosis (**b**) was assessed by evaluating cleaved caspase-3 levels after 24 h exposure (N = 10 repetitions/condition). The blue color indicates DAPI-stained cell nuclei. The red color indicates cleaved caspase-3. Scale bar = 50 µm. Brightness was adjusted in exactly the same way with exactly the same parameters for each condition. Data are presented as mean ± SEM. * *p* < 0.05, *** *p* < 0.001. DOX, doxorubicin. PM, pyridoxamine.

## Data Availability

The data presented in this study are available from the corresponding author on request.

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
