# Peer review of "Pyridoxamine Attenuates Doxorubicin-Induced Cardiomyopathy without Affecting Its Antitumor Effect on Rat Mammary Tumor Cells"

_cells, 2024, doi:10.3390/cells13020120_

Round 1

Reviewer 1 Report

Comments and Suggestions for Authors

The manuscript entitled “Pyridoxamine attenuates doxorubicin-induced cardiomyopathy without affecting its antitumor effect on rat mammary tumor cells” aims to investigate the effect of PM in a rat model of DOX-induced cardiotoxicity and its influence on the DOX antitumor effect in vitro. The idea of this manuscript is impressive. However, it can not be recommended for publication in its current form. I recommend the following revision before its acceptance and publication.

General

The authors can use Grammarly or Ginger software to address minor grammar/spelling errors.

Abstract

Please mention the dose and frequency of DOX and PM. Also, mention the percentage of protection PM provides and other important results.

Materials and methods

1.     The authors are advised to elaborate on section 2.1.

2.     Line 73: Please mention the reason for the dose selection of PM.

3.     Line 74: Does “drinking water” mean PM was administered orally? How much dose was administered, and how many times per day?

4.     Section 2.3: Please mention how the results were obtained for this section.

5.     Section 2.4 and 2.5: Please elaborate on the experimental part. Also, provide the closest reference to carry out a similar experiment.

Results

The authors are advised to summarize the headings/main points of sections 3.1, 3.2, 3.3 and 3.4 as an additional figure without deleting other figures provided in the manuscript.

Discussion

1.     Line 238: “First to report that PM offers cardioprotection in a rat model”. Please compare your study with the PM-based results reported in Toxicology, 2010 Jan 31;268(1-2):89-97. doi: 10.1016/j.tox.2009.12.004.

2.     Line 241-253: The unnecessary content of these lines can be reduced. Keep discussion directed to PM and its close analogs.

3.     Line 272: “Paper submitted”. Please provide a proper reference.

4.     Line 282-289: Please concise this part.

The authors can discuss some aspects related to PM—for example, the effect of pyridoxamine-containing food on reducing DOX induced cardiomyopathy.

Comments on the Quality of English Language

Minor editing of English language required

Reviewer 2 Report

Comments and Suggestions for Authors

The manuscript entitled "Pyridoxamine attenuates doxorubicin-induced cardiomyopathy without affecting its antitumor effect on rat mammary tumor cells" is an original work presenting in vitro data suggesting the possible application of pyridoxamine in reducing some of the adverse effects of doxorubicin. The study is well-structured with proper control and comparison groups. Adequate experimental setups were proposed, especially for for the in vitro cell experiments. The rats were allowed free access to water with pyridoxamine (1 g/L) but it is unclear what is the amount they actually took. The results presented in the current study are promising but definitely would require further and more detail evaluation. The authors have investigated the effects on LA7 mammary tumor cells. What about the possible effects on other cells (e.g. pyridoxamine is known to have toxic effects in higher doses to neuronal cells). What possible route of administration with the simultaneous DOX treatment do the authors expect - parenteral or oral? Furthermore, whether PM will be applied together with DOX or as a separate dosage form. In my opinion, a lot of questions arise from the presented study. But as a communication, the work is well written, with scientific quality and would be a ground for significant future studies.

Reviewer 3 Report

Comments and Suggestions for Authors

The manuscript presented by Haesen et al. presents studies on the effect of pyridoxamine on DOX-induced cardiotoxicity in an in vivo rat model and its effect on the antitumor activity of DOX in vitro in LA7 rat mammary tumor cells.

Unfortunately, in the reviewer's opinion, the research is not suitable for publication in the journal Cells and raises deep doubts about its innovativeness.

First of all, some of the studies performed in the in vivo rat model seem very similar to the studies presented in the publication "Pyridoxamine Protects Against Cardiotoxicity After Doxorubicin Chemotherapy" (DOI: 10.1161/circ.144.suppl_1.10507). What is the difference between the research in these two publications? Both examine the effect of PM on doxorubicin cardiotoxicity in a rat model. Moreover, some of the authors in both manuscripts overlap, and the authors did not refer in any way to the publication cited by the reviewer.

Secondly, studies of the effect of DOX on LA7 cells are not new. The only new thing in the research is the fact that PM does not abolish the therapeutic effect of DOX. Unfortunately, there is not enough research to consider the manuscript as suitable for publication.

Another factor is the fact that the authors declare that they have performed immunocytochemical tests, but they have not attached them to the publication or the supplement.

According to the reviewer, the manuscript does not meet the basic criteria for publication in the journal Cells.

Comments on the Quality of English Language

Moderate editing of English language required

Round 2

Reviewer 3 Report

Comments and Suggestions for Authors

Due to the fact that it is a communication, the reviewer accepts the authors' explanations and recommends the manuscript for publication.